# Curvature effects on phase transitions in chiral magnets

Kostiantyn V. Yershov[1,2⋆], Volodymyr P. Kravchuk[1,3],
Denis D. Sheka[4] and Ulrich K. Rößler[2]

**1** Bogolyubov Institute for Theoretical Physics of National Academy of Sciences of Ukraine,
03143 Kyiv, Ukraine
**2** Leibniz-Institut für Festkörper- und Werkstoffforschung,
IFW Dresden, 01069 Dresden, Germany
**3** Institut für Theoretische Festkörperphysik, Karlsruher Institut für Technologie,
D-76131 Karlsruhe, Germany
**4** Taras Shevchenko National University of Kyiv, 01601 Kyiv, Ukraine

⋆ yershov@bitp.kiev.ua

## Abstract

Periodical equilibrium states of magnetization exist in chiral ferromagnetic films, if the constant of antisymmetric exchange (Dzyaloshinskii–Moriya interaction) exceeds some critical value. Here, we demonstrate that this critical value can be significantly modified in curved film. The competition between symmetric and antisymmetric exchange interactions in a curved film can lead to a new type of domain wall which is inclined with respect to the cylinder axis. The wall structure is intermediate between Bloch and Néel ones. The exact analytical solutions for phase boundary curves and the new domain wall are obtained.


## 1   Introduction

Magnetic nanostructure with arbitrary curvilinear shapes can acquire a multitude of ground-state configurations [1–4] under the twisting influence of the Dzyaloshinskii–Moriya interaction (DMI) and the effect of the curved surfaces/interfaces. Modulated states arise, if a nanomagnet has typical lengths comparable to the twisting length [5–7] which is determined by the material parameters and can be influenced by the curvature. The magnetic phase-diagram of curvilinear ferromagnets becomes much richer as compared to a flat specimen. Among simple curvilinear shapes, hollow cylindrical tubes or wires are very promising for a broad range of biomedical [8–11] and technological [12, 13] applications, also see Review 14. Nanotubes can also be assembled into interconnected networks [15] which makes them attractive for advanced hardware concepts in neuromorphic computing [16]. It is important to note that magnetic nanotubes can be produced experimentally with different techniques [17–22].

    Magnetic nanotubes belong to the simplest magnetic systems with pattern-induced chirality breaking [1, 23]: two energetically equivalent vortex domain walls (DWs) with opposite chiralities possess different dynamical properties, leading to a suppression of the Walker breakdown [24] and Cherenkov-like radiation of magnons for fast DWs [25, 26]. Additionally, tubular geometry results in the asymmetric spin-wave dispersion relation in azimuthally magnetized tubes [27, 28], similarly to systems with intrinsic DMI [29, 30]. In this context, an interrelation between effects due to intrinsic DMI and curvature-induced chirality is expected. An important question is, how the curvature modifies the critical DMI $d_0$ [5, 7], which separates homogeneous and periodic magnetization structures. This is important for assessing the stability of skyrmions [31] and their motion [32] along the tubes and other curvilinear surfaces [33, 34]. Here, we present a detailed study of equilibrium states of the ferromagnetic nanotubes with intrinsic DMI of different symmetries. We show that: (i) The curvature modifies the critical DMI strength. (ii) New types of DWs appear in the periodic phase.

## 2   Model

We consider the tubular shell as a ribbon of thickness $h$ and width $w$, close-coiled upon the rod of radius $R$, see Fig. 1. The central line of the ribbon makes angle $\pi/2 - \psi$ with the cylinder axis. The ribbon width is determined as $w = 2\pi R \sin\psi$, this results in a closed cylindrical surface, i.e. without a bordering rim along the axis. The surface of the ribbon $\varsigma$ can be parameterized in the following way: $\varsigma(x_1, x_2) = R\cos(\rho_s/R)\hat{x} + R\sin(\rho_s/R)\hat{y} + \rho_z\hat{z}$, where $\rho_s = x_1\cos\psi - x_2\sin\psi$ and $\rho_z = x_1\sin\psi + x_2\cos\psi$, $x_1 \in [0, L]$ and $x_2 \in [-w/2, w/2]$ are coordinates within the ribbon surface, see Fig. 1. Such a nontrivial parametrization of the cylinder surface is useful for description of DWs which may be arbitrarily oriented along the

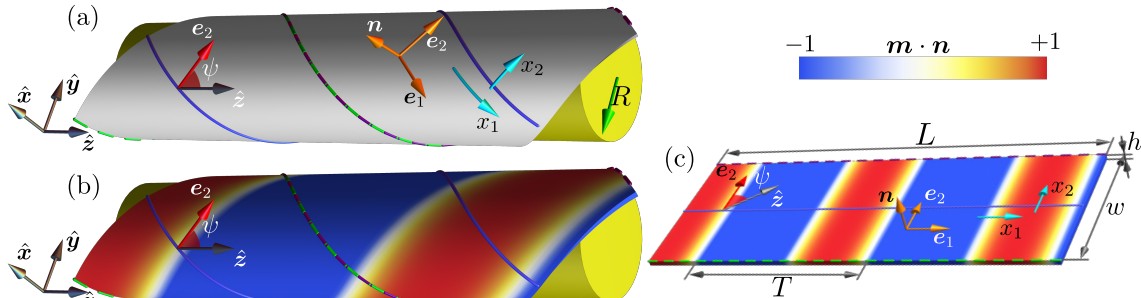

Figure 1: **Schematic presentation of the geometry:** (a),(b) The tubular shell is presented as a ribbon [gray in (a) and colored surfaces in (b)] of thickness $h$ and width $w$ which is tightly (without gaps) rolled up around the rod (yellow) of radius $R$; (c) unrolled ribbon. Thick blue line $\varsigma(x_1, 0)$ corresponds to the ribbon center, dashed lines $\varsigma(x_1, -w/2)$ and $\varsigma(x_1, w/2)$ with $w = 2\pi R \sin\psi$ correspond to the ribbon side edges. Color scheme corresponds to the normal magnetization component.

tube axis, i.e. $\psi$ defines the angle between the DW and $\hat{z}$ axis. Parametrization $\varsigma(x_1, x_2)$ induces the natural tangential basis $\boldsymbol{e}_\alpha = \partial_\alpha \varsigma$ with the corresponding Euclidean metric tensor elements $g_{\alpha\beta} = \boldsymbol{e}_\alpha \cdot \boldsymbol{e}_\beta = \delta_{\alpha\beta}$. Here, $\alpha, \beta = 1, 2$ and $\partial_\alpha \equiv \partial_{x_\alpha}$. Note that in our particular case $\boldsymbol{e}_\alpha$ are orthogonal vectors of unit length. This enables us to introduce the orthonormal basis $\{\boldsymbol{e}_1, \boldsymbol{e}_2, \boldsymbol{n}\}$, where $\boldsymbol{n} = \boldsymbol{e}_1 \times \boldsymbol{e}_2$ is a normal vector to the surface, see Fig. 1.

Assuming small thickness of the coiled film ($h \ll R$), we consider the magnetization as a continuous function of two variables $\boldsymbol{M} = \boldsymbol{M}(x_1, x_2)$, which obeys the periodic boundary condition $\boldsymbol{M}(x_1, w/2) = \boldsymbol{M}(x_1 + T, -w/2)$ with $T = 2\pi R \cos\psi$. Such constraint for $\boldsymbol{M}$ is a requirement of continuity of the magnetization for the used parameterization of the cylinder surface. The energy of the system is modelled by the functional

$$E = h \int \int \left[ \mathscr{A} \mathscr{E}_{\mathrm{X}} + \mathscr{K}\left(1 - m_n^2\right) + \mathscr{D} \mathscr{E}_{\mathrm{D}} \right] \mathrm{d}x_1 \mathrm{d}x_2, \qquad (1)$$

where three contributions are taken into account. The first term in (1) is the exchange energy density with $\mathscr{E}_{\mathrm{X}} = \sum_{i=x,y,z} (\partial_i \boldsymbol{m})^2$, where $\mathscr{A}$ is the exchange constant. Here $\boldsymbol{m} = \boldsymbol{M}/M_s$ is the unit magnetization vector with $M_s$ the saturation magnetization. The second term is the easy-normal anisotropy, where $\mathscr{K} > 0$ and $m_n = \boldsymbol{m} \cdot \boldsymbol{n}$ is the normal magnetization component. The competition between exchange and anisotropy results in the magnetic length $\ell = \sqrt{\mathscr{A}/\mathscr{K}}$, which determines a length scale of the system. The last term in (1) represents DMI contribution $\mathscr{E}_{\mathrm{D}}$ with $\mathscr{D}$ the DMI constant. We consider two types of DMI: (i) $\mathscr{E}_{\mathrm{D}}^{\mathrm{B}} = \boldsymbol{m} \cdot [\boldsymbol{\nabla} \times \boldsymbol{m}]$ is applicable for systems with $T$ and $O$ symmetries [30]. In the following this is called DMI of Bloch type, since for planar films it results in DWs and skyrmions of Bloch type. (ii) $\mathscr{E}_{\mathrm{D}}^{\mathrm{N}} = m_n \boldsymbol{\nabla} \cdot \boldsymbol{m} - \boldsymbol{m} \cdot \boldsymbol{\nabla} m_n$ is valid for ultrathin films [35, 36], bilayers [37] or materials belonging to $C_{nv}$ crystal classes. In the following we call this DMI of Néel type. Here and below the indices B and N correspond to the Bloch and Néel DMI types, respectively.

In our model, we assume that the magnetostatic contribution is negligibly small as compared with the anisotropy contribution, i.e. we consider systems with quality factor $Q = 2\mathscr{K}/\left(\mu_0 M_s^2\right) \gg 1$ [38]. Examples of chiral magnets which satisfies these condition were recently studied in [32, 39]. Additionally, for thin stripes the magnetostatic contribution can be reduced to an effective easy-surface anisotropy [23, 40–43], which simply results in a shift of the anisotropy constant $\mathscr{K} \rightarrow \mathscr{K} - \mu_0 M_s^2/2$. This approximation is widely used for the description of equilibrium states on toroidal nanoshells [44], statics and dynamics of skyrmions [32, 45–47] and DWs [48] in curved nanoshells.

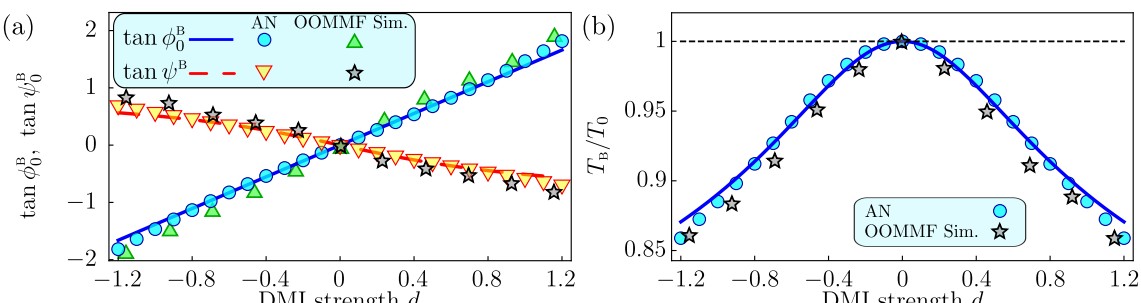

Figure 2: **Parameters of periodical state in tubular shells:** (a) and (b) show the angles $\phi_0^{\text{B}}$, $\psi^{\text{B}}$, and period $T_{\text{B}}$ as functions of DMI for $\varkappa \approx 0.72$. In (a), lines are plotted by means of (3); in (b), solid line is $T_{\text{B}}/T_0 = |\cos\psi^{\text{B}}|$. Symbols correspond to numerical simulations: "AN" – spin-lattice simulations with the effectively reduced anisotropy constant [see Appendix C]; and "OOMMF Sim." – full-scale micromagnetic simulations [see Appendix D].

Using a curvilinear reference frame we parametrize the magnetization in the following way $\boldsymbol{m} = \sin\theta\cos\phi\,\boldsymbol{e}_1 + \sin\theta\sin\phi\,\boldsymbol{e}_2 + \cos\theta\,\boldsymbol{n}$. Expressions for $\mathscr{E}_{\text{X}}$, $\mathscr{E}_{\text{D}}^{\text{B}}$, and $\mathscr{E}_{\text{D}}^{\text{N}}$ for a general case of a local curvilinear basis were previously obtained in Refs. [49], [50], and [45], respectively (also see Appendix A). In the following we look for the equilibrium magnetization states. To this end we minimize energy (1) with respect to functions $\theta(x_1, x_2)$, $\phi(x_1, x_2)$ and constant $\psi$.

# 3 DMI of Bloch type

First, we consider the case of Bloch DMI $\mathscr{E}_{\text{D}} = \mathscr{E}_{\text{D}}^{\text{B}}$. For such kind of DMI we find two solutions, see Appendix B.1. The **homogeneous** (in the curvilinear reference frame) solution corresponds to the hedgehog state ($\boldsymbol{m} = \pm\boldsymbol{n}$), its total energy normalized by $E_0 = hwL\mathscr{K}$ is

$$\mathscr{E}_{\text{B}}^{\text{un}} = \varkappa^2, \tag{2}$$

where $\varkappa = \ell/R$ is the dimensionless curvature. Additionally an **inhomogeneous** solution is found with

$$\tan\phi_0^{\text{B}} = -\tan 2\psi^{\text{B}} = \frac{d}{\varkappa}, \tag{3}$$

where $d = \mathscr{D}/\sqrt{\mathscr{A}\mathscr{K}}$ is DMI strength. It is important that angle $\phi_0^{\text{B}}$, which defines orientation of the tangential magnetization component, is a coordinate independent constant. The relation (3) can be interpreted as follows: for given $d$ and $\varkappa$, there is a curvilinear frame of reference determined by the angle $\psi^{\text{B}}$ in which the magnetization angle $\phi^{\text{B}}$ is constant. Angles $\phi_0^{\text{B}}$ and $\psi^{\text{B}}$ as functions of DMI strength are plotted in Fig. 2(a). For both types of DMI angle $\theta(x_1)$, which defines the magnitude of the normal magnetization component of the inhomogeneous state, depends on only one coordinate $x_1$, oriented along the stripe, see Fig. 1. It is determined by the common "DW" equation

$$\theta_{\xi_1\xi_1} - \lambda\sin\theta\cos\theta = 0, \tag{4}$$

with the solution

$$\theta(\xi_1) = \text{am}\left(\sqrt{C}\,\xi_1, -\frac{\lambda}{C}\right), \tag{5}$$

where $\text{am}(\bullet, \bullet)$ is Jacobi's amplitude [51, 52], $C$ is an integration constant, and $\xi_1 = x_1/\ell$ is the dimensionless coordinate. The solution (5) describes the sequence of DWs oriented along

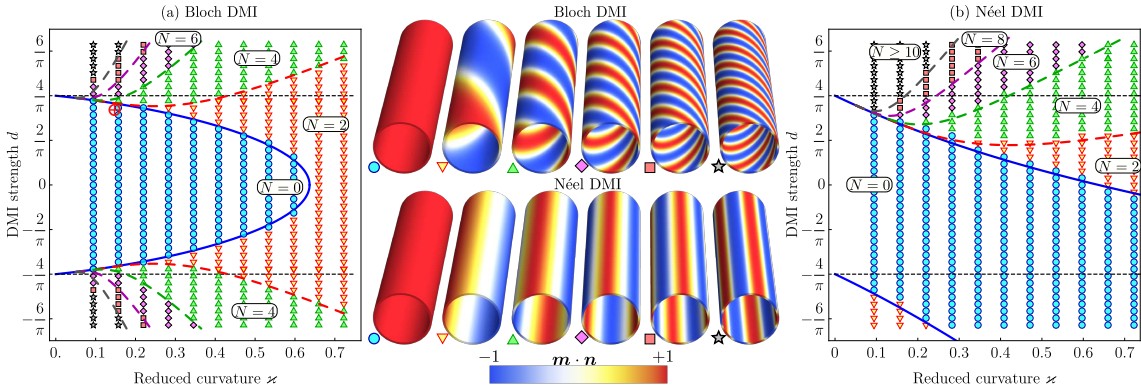

Figure 3: **Equilibrium states in tubular shells:** (a) and (b) show phase diagrams of equilibrium states in tubular shell with Bloch and Néel type of intrinsic DMI, respectively. Symbols display the results from numerical simulations: circles – normal (hedgehog) magnetization distribution ($m = \pm n$); other symbols – periodic states (gray stars correspond to states with $q \geq 5$). Blue solid lines in (a) and (b) are analytical critical lines determined by Eqs. (8) and (12), respectively; dashed lines in (a) and (b) mark transitions, where the periodic equilibrium states change their number of DWs, as determined by numerical solution of energies equality $\mathscr{E}_{\mathrm{B}}^{\mathrm{per}}(q) = \mathscr{E}_{\mathrm{B}}^{\mathrm{per}}(q+1)$ and $\mathscr{E}_{\mathrm{N}}^{\mathrm{per}}(q) = \mathscr{E}_{\mathrm{N}}^{\mathrm{per}}(q+1)$: $q = 1$ corresponds to red dashed line, $q = 2$ – green, $q = 3$ – purple, $q = 4$ – gray. Dashed black horizontal lines correspond to critical DMI parameter in a flat systems $d_0 = \pm 4/\pi$. Symbol $\oplus$ in (a) corresponds to the boundary between the hedgehog and periodic states obtained by means of micromagnetic simulations in Ref. [32].

the $x_2$ coordinate (perpendicularly to the ribbon, see Fig. 1). For each type of DMI, parameter $\lambda = \lambda(\varkappa, d)$ is a function of curvature and DMI strength. For well separated DWs, $\lambda$ defines the DW width $\Delta = 1/\sqrt{\lambda}$. The integration constant $C$ determines the period $\theta(\xi_1 + T) = \theta(\xi_1)$

$$T = \frac{4}{\sqrt{C}}\mathrm{K}\left(-\frac{\lambda}{C}\right) = T_0 |\cos\psi|/q, \tag{6}$$

with $\mathrm{K}(\bullet)$ is the complete elliptic integral of the first kind [51, 52]. On the other hand, period $T = T_0 |\cos\psi|/q$ is predetermined by the periodical boundary conditions discussed above. Here $q \in \mathbb{N}$ determines the number of DWs $N = 2q$ on the tube and $T_0 = 2\pi/\varkappa$. $N$ is even due to the periodical boundary conditions enforced by the tubular geometry. For the case of Bloch DMI constant $C \equiv C_{\mathrm{B}}$ is determined by the equation (6) with $\psi = \psi^{\mathrm{B}}$ taken from (3) and $\lambda \equiv \lambda_{\mathrm{B}} = 1 + \varkappa\left(\sqrt{d^2 + \varkappa^2} - \varkappa\right)/2$. One should note that for the case $\varkappa > 0$ and $d \neq 0$ the DW width decreases as compared to the case $\varkappa = 0$ (planar film) or $d = 0$. For the corresponding period we use the notation $T \equiv T_{\mathrm{B}}$. Period $T_{\mathrm{B}}$ as a function of the DMI strength is plotted in Fig. 2(b). The normalized energy of periodic states per period ($T = T_{\mathrm{B}}$) is

$$\mathscr{E}_{\mathrm{B}}^{\mathrm{per}} = \mathscr{E}_{\mathrm{B}}^{\mathrm{un}} + \frac{\varkappa q}{\cos\psi^{\mathrm{B}}}\left[\frac{4}{\pi}\sqrt{C_{\mathrm{B}}(q)}\mathrm{E}\left(-\frac{\lambda_{\mathrm{B}}}{C_{\mathrm{B}}(q)}\right) - \left(\varkappa + \sqrt{\varkappa^2 + d^2}\right) - C_{\mathrm{B}}(q)\frac{\cos\psi^{\mathrm{B}}}{q\varkappa}\right], \tag{7}$$

where $\mathrm{E}(\bullet)$ is the complete elliptic integral of the second kind [51, 52]. For a planar film, the transition between the homogeneous and periodical state is characterized by infinite increase of period of the spiral state [7]. Although, for the cylindrical surface the period is finite in the transition point, for the limit case $\varkappa \to 0$ one has $T \to \infty$. Using that $C \to 0$ in this limit, we obtain from the equality $\mathscr{E}_{\mathrm{B}}^{\mathrm{per}} = \mathscr{E}_{\mathrm{B}}^{\mathrm{un}}$ the analytical expression for the critical DMI

$$d_c^{\mathrm{B}} = \pm d_0 \sqrt{1 - \varkappa^2 - \frac{\varkappa}{2}\frac{\pi^2 - 4}{\pi^2}\left[\varkappa + \sqrt{\varkappa^2 + \pi^2(1 - \varkappa^2)}\right]}, \tag{8}$$

where $d_0 = 4/\pi$ is a critical DMI parameter for flat systems, which separates homogeneous and periodic magnetization distributions [5,7]. Although the expression (8) is obtained in the small curvature limit, it describes very well the existence region of the homogeneous state for the whole range of curvatures, see Fig. 3(a). The boundary (8) is also in a good agreement with results obtained by means of micromagnetic simulations in Ref. [32], see symbol ⊕ in Fig. 3(a). The equality of energies $\mathscr{E}_{\mathrm{B}}^{\mathrm{per}}(\varkappa, d_c, q) = \mathscr{E}_{\mathrm{B}}^{\mathrm{per}}(\varkappa, d_c, q+1)$ determines the boundary between states with different number of DWs. The resulting phase diagram is plotted in Fig. 3(a). In the limit case of very small curvature ($\varkappa \ll 1$), the boundary curve (8) has the asymptotic behavior $d_c^{\mathrm{B}} \approx \pm d_0 \mp \left(1 - 4/\pi^2\right)\varkappa$. Thus the curvature decreases the critical magnitude of the DMI strength. The boundary curve (8) intersects the abscissa with $\varkappa_0^{\mathrm{B}} = 2/\pi$. For $\varkappa > \varkappa_0^{\mathrm{B}}$ the periodical state with two DWs exists even without intrinsic DMI, see Fig. 3. This effect is analogous to the effect of spontaneous formation of the onion state in nanorings when curvature exceeds some critical value [53].

## 4 DMI of Néel type

Let us now consider the case of Néel DMI $\mathscr{E}_{\mathrm{D}} = \mathscr{E}_{\mathrm{D}}^{\mathrm{N}}$. The energy of the **homogeneous** hedgehog state ($\boldsymbol{m} = \pm \boldsymbol{n}$) reads

$$\mathscr{E}_{\mathrm{N}}^{\mathrm{un}} = \varkappa(d + \varkappa). \tag{9}$$

Similarly to the case of Bloch DMI, there is an **inhomogeneous** solution in form of periodical modulation. As well as in the previous case, the angle $\phi$ takes the constant value (for details see Appendix B.2):

$$\cos \phi_0^{\mathrm{N}} = -\mathrm{sgn}(d + 2\varkappa). \tag{10}$$

However, in contrast to the previous case, DWs are always aligned along the cylinder. This corresponds to the equilibrium value $\psi^{\mathrm{N}} = 0$ (or equivalently $\psi^{\mathrm{N}} = \pi$). As previously, the normal magnetization component is described by the same Eq. (5) with $C \equiv C_{\mathrm{N}}$ determined by (6) with $\psi = \psi^{\mathrm{N}}$ and $\lambda \equiv \lambda_{\mathrm{N}} = 1 - \varkappa d$. Note that due to non-zero DMI and curvature the width of the well separated Néel DWs is increased. This behavior is opposite to the case of the Bloch DWs. The normalized energy of the modulated state per period ($T = T_{\mathrm{N}} = T_0/q$) is

$$\mathscr{E}_{\mathrm{N}}^{\mathrm{per}} = \mathscr{E}_{\mathrm{N}}^{\mathrm{un}} + \varkappa q\left[\frac{4}{\pi}\sqrt{C_{\mathrm{N}}(q)}\mathrm{E}\left(-\frac{\lambda_{\mathrm{N}}}{C_{\mathrm{N}}(q)}\right) - |2\varkappa + d| - \frac{C_{\mathrm{N}}(q)}{q\varkappa}\right]. \tag{11}$$

The equality of energies $\mathscr{E}_{\mathrm{N}}^{\mathrm{un}}(\varkappa, d_c) = \mathscr{E}_{\mathrm{N}}^{\mathrm{per}}(\varkappa, d_c)$ determines the boundary between homogeneous and periodic states. In the small curvature limit we obtain

$$d_c^{\mathrm{N}} = \pm d_0\left[\sqrt{1 + 2\varkappa^2\left(1 + \frac{2}{\pi^2}\right)} \mp \varkappa\left(\frac{2}{\pi} + \frac{\pi}{2}\right)\right]. \tag{12}$$

As in the case of Bloch type DMI, the expression (12) describes the boundary of the homogeneous state in the phase diagram for a wide range of curvatures. The equality of energies $\mathscr{E}_{\mathrm{N}}^{\mathrm{per}}(\varkappa, d_c, q) = \mathscr{E}_{\mathrm{N}}^{\mathrm{per}}(\varkappa, d_c, q+1)$ determines the boundary between states with different number of DWs. The resulting phase diagram is plotted in Fig. 3(b). In the limit case of very small curvature ($\varkappa \ll 1$), the boundary curve (12) has the linear asymptotic behavior $d_c^{\mathrm{N}} \approx \pm d_0 - 2\left(1 + 4/\pi^2\right)\varkappa$. Thus, due to the curvature the absolute value of the critical DMI can be decreased as well as increased depending on the sign of the DMI. Similarly to the case of Bloch type DMI, the boundary curve (12) intersects the abscissa with $\varkappa_0^{\mathrm{N}} = \varkappa_0^{\mathrm{B}} = 2/\pi$ and for the case $\varkappa > \varkappa_0^{\mathrm{N}}$ the periodical state exists even without intrinsic DMI, see Fig. 3.

## 5 Conclusions

At the example of cylindrical thin tubes, we show that curvature modifies the value of critical DMI for curved systems, see Eqs. (8) and (12), which separates the hedgehog state with homogeneous magnetization normal to the film from the inhomogenous modulated states. For the case of Néel type of DMI this effect is much stronger (in the limit case $\varkappa \ll 1$) as compared to the case of the Bloch DMI. The curvature effects are more pronounced for the case of Néel intrinsic DMI because the curvature-induced DMI is usually of the Néel type, thus a direct competition takes place. Note, that for the same reason the Néel skyrmions are more strongly affected by the curvature gradients as compared to the Bloch skyrmions [33] and the DMI-free skyrmions stabilized by curvature are of Néel type [45]. We found an exact solution for equilibrium states on the cylindrical surface for two different types of DMI and plotted the corresponding phase diagrams, see Fig. 3. The presence of the Néel DMI does not modify the structure of DWs, i.e. DWs are oriented along the cylinder axis ($\psi^{\mathrm{N}} = 0$) and they are of Néel type. For the case of Bloch DMI, the DWs are of a type intermediate between Bloch and Néel due to competition of intrinsic DMI and geometry-induced DMI of Néel type. These DWs are inclined by the angle $\psi^{\mathrm{B}} \in (-\pi/4; \pi/4)$, see Eq. (3) and Fig. 3. The direction of DWs inclination (sign of the angle $\psi^{\mathrm{B}}$) is defined by the sign of the DMI parameter. This effect is similar (i) to the field-induced inclined DWs in flat stripes [54]. In our case the role of the external field is played by the geometry-induced easy-axis anisotropy along the cylinder axis. And it also resembles (ii) the DMI-induced chiral twist of domains separated by the head-to-head (tail-to-tail) DWs in nanotubes [48]. In both cases, the periodical boundary conditions, enforced by the closed cylindrical geometry, result in even number of domains on the cylinder.

## Acknowledgements

We thank U. Nitzsche for technical support.

**Funding information**    K.V.Y. acknowledges financial support from UKRATOP-project (funded by BMBF under reference 01DK18002). In part, this work was supported by the Program of Fundamental Research of the Department of Physics and Astronomy of the National Academy of Sciences of Ukraine (Project No. 0120U100855), by the Alexander von Humboldt Foundation (Research Group Linkage Programme), and by Taras Shevchenko National University of Kyiv (Project No. 19BF052-01).

## A    Introduction of the curvilinear basis and magnetic interactions on a curvilinear shell

The surface parametrization $\varsigma(x_1, x_2)$ induces the natural tangential basis $\boldsymbol{g}_\alpha = \partial_\alpha \varsigma$ with the corresponding metric tensor elements $g_{\alpha\beta} = \boldsymbol{g}_\alpha \cdot \boldsymbol{g}_\beta$. Here, $\alpha, \beta = 1, 2$ and $\partial_\alpha \equiv \partial_{x_\alpha}$. As the vectors $\boldsymbol{g}_\alpha$ are orthogonal, one can introduce the orthonormal basis $\{\boldsymbol{e}_1, \boldsymbol{e}_2, \boldsymbol{n}\}$ with

$$\boldsymbol{e}_\alpha = \frac{\boldsymbol{g}_\alpha}{\sqrt{g_{\alpha\alpha}}}, \quad \boldsymbol{n} = \boldsymbol{e}_1 \times \boldsymbol{e}_2. \tag{A.1}$$

Using the Gauß-Godazzi formula and Weingarten's equation [55, 56] one can obtain the following differential properties of the basis vectors

$$\nabla_\alpha \boldsymbol{e}_\beta = h_{\alpha\beta} \boldsymbol{n} - \Omega_\alpha \epsilon_{\beta\gamma} \boldsymbol{e}_\gamma, \quad \nabla_\alpha \boldsymbol{n} = -h_{\alpha\beta} \boldsymbol{e}_\beta. \tag{A.2}$$

Here, $\nabla_\alpha \equiv (g_{\alpha\alpha})^{-1/2} \partial_\alpha$ (no summation over $\alpha$) are components of the surface del operator and $\|h_{\alpha\beta}\|$ is a modified second fundamental form. The second fundamental form determines the Gauß curvature $\mathscr{K} = \det\|h_{\alpha\beta}\|$ and the mean curvature $\mathscr{H} = \mathrm{tr}\|h_{\alpha\beta}\|$. Components of the spin connection vector $\boldsymbol{\Omega}$ are determined by the relation $\Omega_\gamma = \dfrac{1}{2}\epsilon_{\alpha\beta} \boldsymbol{e}_\alpha \cdot \nabla_\gamma \boldsymbol{e}_\beta$.

Using curvilinear reference frame (A.1), we introduce the following magnetization parametrization

$$\boldsymbol{m} = \sin\theta\,\boldsymbol{\varepsilon} + \cos\theta\,\boldsymbol{n}, \quad \boldsymbol{\varepsilon} = \cos\phi\,\boldsymbol{e}_1 + \sin\phi\,\boldsymbol{e}_2, \tag{A.3}$$

where $\theta$ and $\phi$ are magnetic angles, and $\boldsymbol{\varepsilon}$ is a normalized projection of the vector $\boldsymbol{m}$ on the tangential plane.

The first term in (1) is the exchange density $\mathscr{E}_{\mathrm{x}} = \sum_{i=x,y,z}(\partial_i \boldsymbol{m})^2$ with $\mathscr{A}$ the exchange constant. In the curvilinear reference frame exchange energy can be written as [45, 49, 53]

$$\begin{aligned}
\mathscr{E}_{\mathrm{x}} =\ & \nabla_\alpha m_\beta \nabla_\alpha m_\beta + \nabla_\alpha m_n \nabla_\alpha m_n \\
& + 2h_{\alpha\beta}\left(m_\beta \nabla_\alpha m_n - m_n \nabla_\alpha m_\beta\right) + 2\epsilon_{\alpha\beta}\Omega_\gamma m_\beta \nabla_\gamma m_\alpha \\
& + \left(h_{\alpha\gamma}h_{\gamma\beta} + \Omega^2 \delta_{\alpha\beta}\right)m_\alpha m_\beta + \left(\mathscr{H}^2 - 2\mathscr{K}\right)m_n^2 + 2\epsilon_{\alpha\gamma}h_{\gamma\beta}\Omega_\beta m_\alpha m_n.
\end{aligned} \tag{A.4a}$$

Using the angular parametrization (A.3) one can obtain [45, 49, 53]

$$\mathscr{E}_{\mathrm{x}} = [\boldsymbol{\nabla}\theta - \boldsymbol{\Gamma}]^2 + \left[\sin\theta\,(\boldsymbol{\nabla}\phi - \boldsymbol{\Omega}) - \cos\theta\,\partial_\phi \boldsymbol{\Gamma}\right]^2, \tag{A.4b}$$

where $\boldsymbol{\Gamma} = \|h_{\alpha\beta}\| \cdot \boldsymbol{\varepsilon}$.

The second term in (1) corresponds to the Dzyaloshinskii–Moriya interaction (DMI) $\mathscr{E}_{\mathrm{D}}$, with $\mathscr{D}$ being the DMI constant. In the curvilinear frame of reference the Néel type DMI $\mathscr{E}_{\mathrm{D}}^{\mathrm{N}} = m_n \boldsymbol{\nabla}\cdot\boldsymbol{m} - \boldsymbol{m}\cdot\boldsymbol{\nabla}m_n$ can be written as [45]

$$\mathscr{E}_{\mathrm{D}}^{\mathrm{N}} = m_n \nabla_\alpha m_\alpha - m_\alpha \nabla_\alpha m_n - \epsilon_{\alpha\beta}\Omega_\beta m_\alpha m_n - \mathscr{H} m_n^2. \tag{A.5a}$$

Using the angular parametrization (A.3) one can obtain (up to the boundary terms) [45, 46]

$$\mathscr{E}_{\mathrm{D}}^{\mathrm{N}} = 2(\boldsymbol{\nabla}\theta \cdot \boldsymbol{\varepsilon})\sin^2\theta - \mathscr{H}\cos^2\theta + \text{boundary terms}, \tag{A.5b}$$

while, for the Bloch type DMI symmetry $\mathscr{E}_{\mathrm{D}}^{\mathrm{B}} = \boldsymbol{m}\cdot[\boldsymbol{\nabla}\times\boldsymbol{m}]$ this interaction in the curvilinear reference frame reads as [50]

$$\mathscr{E}_{\mathrm{D}}^{\mathrm{B}} = \epsilon_{\alpha\beta}\left(m_n \nabla_\alpha m_\beta - m_\beta \nabla_\alpha m_n\right) + \epsilon_{\alpha\beta}h_{\beta\gamma}m_\alpha m_\gamma - \Omega_\alpha m_\alpha m_n. \tag{A.5c}$$

Substituting the angular parametrization (A.3) into (A.5c) results in the expression (up to the boundary terms) [50]

$$\mathscr{E}_{\mathrm{D}}^{\mathrm{B}} = \sin^2\theta\left[(2\boldsymbol{\nabla}\theta - \boldsymbol{\Gamma})\times\boldsymbol{\varepsilon}\right]\cdot\boldsymbol{n}. \tag{A.5d}$$

The last term in (1) corresponds to the uniaxial anisotropy $\mathscr{E}_{\mathrm{A}} = \sin^2\theta$, with $\mathscr{K} > 0$ the easy-normal anisotropy constant.

Parameterization $\boldsymbol{\varsigma}(x_1, x_2) = R\cos(\rho_s/R)\hat{\boldsymbol{x}} + R\sin(\rho_s/R)\hat{\boldsymbol{y}} + \rho_z\hat{\boldsymbol{z}}$ results in the following first and modified second fundamental forms

$$g_{\alpha\beta} = \delta_{\alpha\beta}, \quad \|h_{\alpha\beta}\| = \frac{1}{R}\left\|\begin{matrix} -\cos^2\psi & \cos\psi\sin\psi \\ \cos\psi\sin\psi & -\sin^2\psi \end{matrix}\right\|, \tag{A.6}$$

respectively. Tubular geometry has zero Gauß curvature $\mathscr{K} = 0$, nonzero mean curvature $\mathscr{H} = -R^{-1}$ (here minus is related to the direction of the normal vector), and zero components of spin connection vector $\boldsymbol{\Omega} = \boldsymbol{0}$.

# B  DMI induced periodical solution for a cylindrical surface

## B.1  DMI of Bloch type

In this section we consider DMI in form $\mathscr{E}_{\mathrm{D}} = \mathscr{E}_{\mathrm{D}}^{\mathrm{B}}$ which is defined in (A.5d). The total energy density in (1) reads as

$$
\begin{aligned}
\frac{\mathscr{E}}{\mathscr{K}} = {}&\left(\tilde{\boldsymbol{\nabla}}\theta\right)^2 + \left(\tilde{\boldsymbol{\nabla}}\phi\right)^2 \sin^2\theta + 2\varkappa \cos\left(\phi+\psi\right)\tilde{\boldsymbol{\nabla}}\theta\cdot\boldsymbol{\eta} \\
&-2\varkappa \sin\theta\cos\theta\sin\left(\phi+\psi\right)\tilde{\boldsymbol{\nabla}}\phi\cdot\boldsymbol{\eta} + \varkappa^2\left[1-\sin^2\theta\sin^2\left(\phi+\psi\right)\right] + \sin^2\theta \\
&+d\sin^2\theta\left[2\left(\theta_{\xi_1}\sin\phi - \theta_{\xi_2}\cos\phi\right) + \frac{\kappa}{2}\sin 2\left(\phi+\psi\right)\right], \quad \boldsymbol{\eta} = \boldsymbol{e}_1\cos\psi - \boldsymbol{e}_2\sin\psi,
\end{aligned}
\tag{B.1}
$$

where $\varkappa = \ell/R$ is a reduced curvature with $\ell = \sqrt{\mathscr{A}/\mathscr{K}}$ being the magnetic length, the operator $\tilde{\boldsymbol{\nabla}}$ acts on the dimensionless curvilinear coordinates $\xi_\alpha = x_\alpha/\ell$, and $d = \mathscr{D}/\sqrt{\mathscr{A}\mathscr{K}}$ is a reduced DMI strength. The equilibrium values of $\theta$, $\phi$, and $\psi$ are determined by the equations

$$
\begin{aligned}
\frac{1}{2}\frac{\delta E}{\delta\theta} = {}&-\tilde{\Delta}\theta + \sin\theta\cos\theta\left[\left(\tilde{\boldsymbol{\nabla}}\phi\right)^2 + 1\right] + 2\varkappa\sin^2\theta\sin\left(\phi+\psi\right)\tilde{\boldsymbol{\nabla}}\phi\cdot\boldsymbol{\eta} \\
&-\varkappa^2\sin\theta\cos\theta\sin^2\left(\phi+\psi\right) - d\left[\sin^2\theta\,\tilde{\boldsymbol{\nabla}}\phi\cdot\boldsymbol{\varepsilon} - \frac{1}{4}\varkappa\sin 2\theta\sin 2\left(\phi+\psi\right)\right] = 0, \\
\frac{1}{2}\frac{\delta E}{\delta\phi} = {}&-\tilde{\boldsymbol{\nabla}}\cdot\left[\sin^2\theta\,\tilde{\boldsymbol{\nabla}}\phi\right] - 2\varkappa\sin\left(\phi+\psi\right)\sin^2\theta\,\tilde{\boldsymbol{\nabla}}\theta\cdot\boldsymbol{\eta} - \varkappa^2\sin^2\theta\sin\left(\phi+\psi\right)\cos\left(\phi+\psi\right) \\
&+d\sin^2\theta\left[\tilde{\boldsymbol{\nabla}}\theta\cdot\boldsymbol{\varepsilon} + \frac{\varkappa}{2}\cos 2\left(\phi+\psi\right)\right] = 0, \\
\frac{1}{2\varkappa}\frac{\delta E}{\delta\psi} = {}&-\theta_{\xi_1}\sin\left(\phi+2\psi\right) - \theta_{\xi_2}\cos\left(\phi+2\psi\right) - \frac{1}{2}\sin 2\theta\left[\phi_{\xi_1}\cos\left(\phi+2\psi\right) - \phi_{\xi_2}\sin\left(\phi+2\psi\right)\right] \\
&-\varkappa\sin^2\theta\sin\left(\phi+\psi\right)\cos\left(\phi+\psi\right) + \frac{d}{2}\sin^2\theta\cos 2\left(\phi+\psi\right) = 0.
\end{aligned}
\tag{B.2}
$$

Here $\tilde{\nabla}_\alpha = \ell\nabla_\alpha \equiv \partial_{\xi_\alpha}$ and $\tilde{\Delta} = \tilde{\nabla}^2$.

Equations (B.2) have a trivial solutions $\theta \equiv 0$ and $\theta \equiv \pi$, which corresponds to the uniform magnetization distribution in the curvilinear reference frame, i.e. $\boldsymbol{m} = \pm\boldsymbol{n}$, with energy (2).

We also found an inhomogeneous solution (3) with $\phi = \phi_0^{\mathrm{B}} = \mathrm{const}$:

$$
\cos\phi_0^{\mathrm{B}} = -\frac{\varkappa\,\mathrm{sgn}(d)}{\sqrt{\varkappa^2 + d^2}}, \quad \sin\psi^{\mathrm{B}} = -\frac{d}{\sqrt{2\left[d^2 + \varkappa\left(\varkappa + \sqrt{\varkappa^2 + d^2}\right)\right]}}.
\tag{B.3}
$$

Note that for the case $d > 0$ one has $-\pi/4 \le \psi^{\mathrm{B}} \le 0$ and $\phi_0^{\mathrm{B}} = \pi - 2\psi^{\mathrm{B}}$. While for the case $d < 0$ one has $0 \le \psi^{\mathrm{B}} \le \pi/4$ and $\phi_0^{\mathrm{B}} = -2\psi^{\mathrm{B}}$. The magnetic angle $\theta$ is defined by the equation (4) with the solution (5).

Energy as a function of DMI strength for Bloch DMI for different $q$ is plotted in Fig. 4(a).

## B.2  DMI of Néel type

Here we consider DMI in form $\mathscr{E}_{\mathrm{D}} = \mathscr{E}_{\mathrm{D}}^{\mathrm{N}}$ which is defined in Eqs. (A.5a) and (A.5b). The total energy density in (1) reads as

$$
\begin{aligned}
\frac{\mathscr{E}}{\mathscr{K}} = {}&\left(\tilde{\boldsymbol{\nabla}}\theta\right)^2 + \left(\tilde{\boldsymbol{\nabla}}\phi\right)^2\sin^2\theta + 2\varkappa\cos\left(\phi+\psi\right)\tilde{\boldsymbol{\nabla}}\theta\cdot\boldsymbol{\eta} - 2\varkappa\sin\theta\cos\theta\sin\left(\phi+\psi\right)\tilde{\boldsymbol{\nabla}}\phi\cdot\boldsymbol{\eta} \\
&+\varkappa^2\left[1-\sin^2\theta\sin^2\left(\phi+\psi\right)\right] + d\left[2\left(\tilde{\boldsymbol{\nabla}}\theta\cdot\boldsymbol{\varepsilon}\right)\sin^2\theta + \varkappa\cos^2\theta\right] + \sin^2\theta.
\end{aligned}
\tag{B.4}
$$

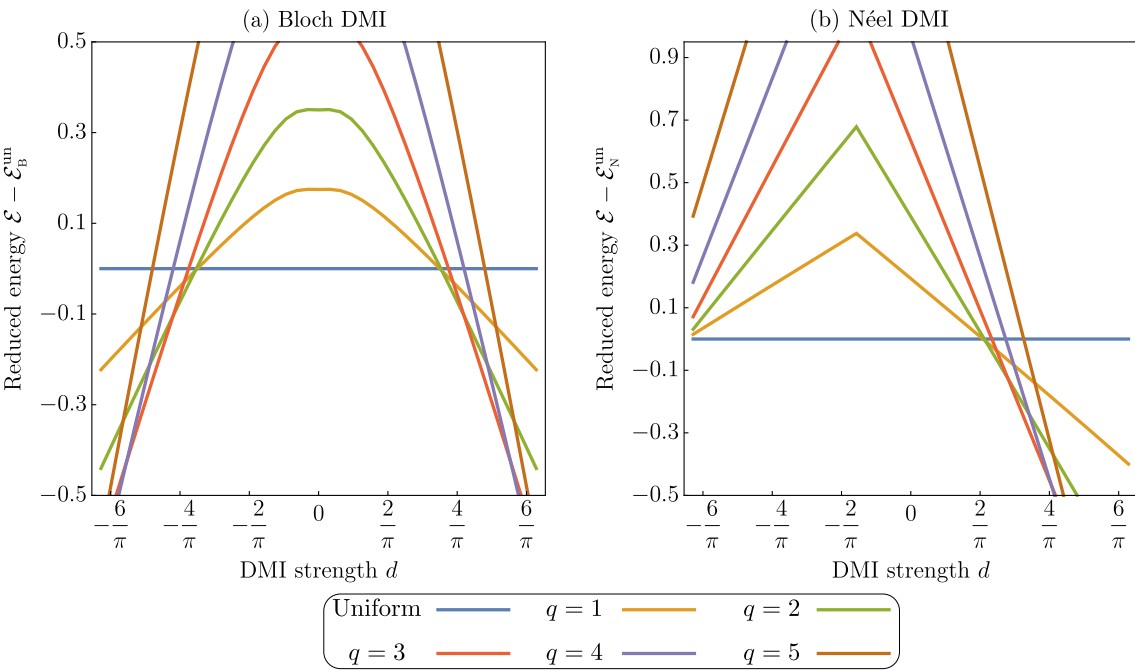

Figure 4: Energies of the cylinder with $\varkappa = 0.25$ for Bloch (a) and Néel (b) DMI types.

The equilibrium values of $\theta$, $\phi$, and $\psi$ determined by the equations

$$\frac{1}{2}\frac{\delta E}{\delta\theta} = -\tilde{\Delta}\theta + \sin\theta\cos\theta\left[\left(\tilde{\boldsymbol{\nabla}}\phi\right)^2 + 1\right] + 2\varkappa\sin^2\theta\sin(\phi+\psi)\tilde{\boldsymbol{\nabla}}\phi\cdot\boldsymbol{\eta}$$

$$-\varkappa^2\sin\theta\cos\theta\sin^2(\phi+\psi) - d\left[\sin^2\theta\,\tilde{\boldsymbol{\nabla}}\phi\cdot\frac{\partial\boldsymbol{\varepsilon}}{\partial\phi} + \varkappa\sin\theta\cos\theta\right] = 0,$$

$$\frac{1}{2}\frac{\delta E}{\delta\phi} = -\tilde{\boldsymbol{\nabla}}\cdot\left[\sin^2\theta\,\tilde{\boldsymbol{\nabla}}\phi\right] - 2\varkappa\sin(\phi+\psi)\sin^2\theta\,\tilde{\boldsymbol{\nabla}}\theta\cdot\boldsymbol{\eta} - \varkappa^2\sin^2\theta\sin(\phi+\psi)\cos(\phi+\psi)$$

$$+d\sin^2\theta\,\tilde{\boldsymbol{\nabla}}\theta\cdot\frac{\partial\boldsymbol{\varepsilon}}{\partial\phi} = 0,$$

$$\frac{1}{2\varkappa}\frac{\delta E}{\delta\psi} = -\theta_{\xi_1}\sin(\phi+2\psi) - \theta_{\xi_2}\cos(\phi+2\psi) - \frac{1}{2}\sin 2\theta\left[\phi_{\xi_1}\cos(\phi+2\psi) - \phi_{\xi_2}\sin(\phi+2\psi)\right]$$

$$-\varkappa\sin^2\theta\sin(\phi+\psi)\cos(\phi+\psi) = 0.$$

(B.5)

Equations (B.5) have a trivial solutions $\theta \equiv 0$ and $\theta \equiv \pi$, which corresponds to the hedgehog state $\boldsymbol{m} = \pm\boldsymbol{n}$, i. e. the homogeneous magnetization distribution in the curvilinear reference frame; the energy of the hedgehog state is described by Eq. (9).

We also found an inhomogeneous solution with $\phi = \phi_0^{\mathrm{N}} = $ const, see (10), and magnetic angle $\theta$ defined in (5).

Energy as a function of DMI strength for Néel DMI for different $q$ is plotted in Fig. 4(b).

## C  Details of the spin-lattice simulations

In order to verify our analytical calculations we perform a set numerical simulations for a ferromagnetic cylindrical surface. We consider a cylindrical surface as a square lattice with lattice constant $a$. Each node is characterized by a magnetic moment $\boldsymbol{m_p}(t)$ which is located

at the position $r_p(t)$. Here $p = (i, j)$ is a two dimensional vector which defines the magnetic moment and its position on the lattice with size $N_1 \times N_2$ ($i \in [1, N_1]$ and $j \in [1, N_2]$). Magnetic moments are ferromagnetically coupled. We are interested in the case when the system is a closed cylindrical surface, hence we impose the periodical boundary conditions $m_{(N_1+1, j)} = m_{(1, j)}$ and $r_{(N_1+1, j)} = r_{(1, j)}$. The dynamics of magnetic system is govern by discrete Landau–Lifshitz–Gilbert equations

$$\frac{\mathrm{d}m_p}{\mathrm{d}\tau} = m_p \times \frac{\partial \mathcal{H}}{\partial m_p} + \alpha m_p \times \left[ m_p \times \frac{\partial \mathcal{H}}{\partial m_p} \right], \tag{C.1}$$

where $\tau = \omega_0 t$ is a reduced time with $\omega_0 = \mu_0 |\gamma_0| M_s$, $\alpha$ is a dimensionless damping coefficients, and $\mathcal{H}$ is a dimensionless energy normalized by $\mu_0 M_s^2$. We consider four contributions to the energy of the system

$$\mathcal{H} = \mathcal{H}_\mathrm{X} + \mathcal{H}_\mathrm{A} + \mathcal{H}_\mathrm{D} + \mathcal{H}_\mathrm{DDI}. \tag{C.2a}$$

The first term in (C.2a) is the exchange energy

$$\mathcal{H}_\mathrm{X} = -\frac{1}{2} \frac{\ell_\mathrm{x}^2}{a^2} \sum_{p, \delta} m_p \cdot m_{p+\delta}, \tag{C.2b}$$

where $\delta$ runs over nearest neighbours of the square lattice and $\ell_\mathrm{x} = \sqrt{\mathcal{A}/(\mu_0 M_s^2)}$.

The second term in (C.2a) is the anisotropy energy

$$\mathcal{H}_\mathrm{A} = -\frac{Q}{2} \sum_p (m_p \cdot n_p)^2, \tag{C.2c}$$

where $n_p$ is easy-normal axis vector at node with coordinate $r_p$, and $Q = 2\mathcal{K}/(\mu_0 M_s^2)$ is a quality factor [38].

The third term in Eq. (C.2a) is a DMI energy

$$\mathcal{H}_\mathrm{D} = \frac{d}{2} \frac{\ell_\mathrm{x}}{a} \sqrt{\frac{Q-\Lambda}{2}} \sum_{p, \delta} d_{p, \delta} \cdot \left[ m_p \times m_{p+\delta} \right], \tag{C.2d}$$

where $d_{p, \delta}$ is a DMI vector. For the case of Néel DMI $d_{p, \delta} = n_p \times u_{p, \delta}$ with $u_{p, \delta} = (r_{p+\delta} - r_p)/a$ being a unit vector which connects two nearest neighbors. For Bloch DMI symmetry we have $d_{p, \delta} = u_{p, \delta}$. Parameter $\Lambda = \{0, 1\}$ defines whether long range dipole-dipole interaction is present or not, i.e. $\Lambda = 0$ corresponds to simulations without dipole-dipole interaction and $\Lambda = 1$ vice versa.

The last term in Eq. (C.2a) is a long range dipole-dipole interaction

$$\mathcal{H}_\mathrm{DDI} = \Lambda \frac{a^3}{8\pi} \sum_{\substack{p, b \\ p \neq b}} \left[ \frac{m_p \cdot m_b}{|r_{pb}|^3} - 3 \frac{(m_p \cdot r_{pb})(m_b \cdot r_{pb})}{|r_{pb}|^5} \right], \tag{C.2e}$$

where $r_{pb} = r_p - r_b$.

For analytical calculations the dipole-dipole effects can be approximated by a simple redefinition of the anisotropy constants, leading to a new magnetic length,

$$\mathcal{K} \to \mathcal{K}^\mathrm{eff} = \mathcal{K} - \Lambda \mu_0 M_s^2/2,$$

$$\ell \to \ell^\mathrm{eff} = \sqrt{\frac{\mathcal{A}}{\mathcal{K}^\mathrm{eff}}} = \ell_\mathrm{x} \sqrt{\frac{2}{Q-\Lambda}}, \tag{C.3}$$

$$d \to d^\mathrm{eff} = \frac{\mathcal{D}}{\sqrt{\mathcal{A}\mathcal{K}^\mathrm{eff}}}.$$

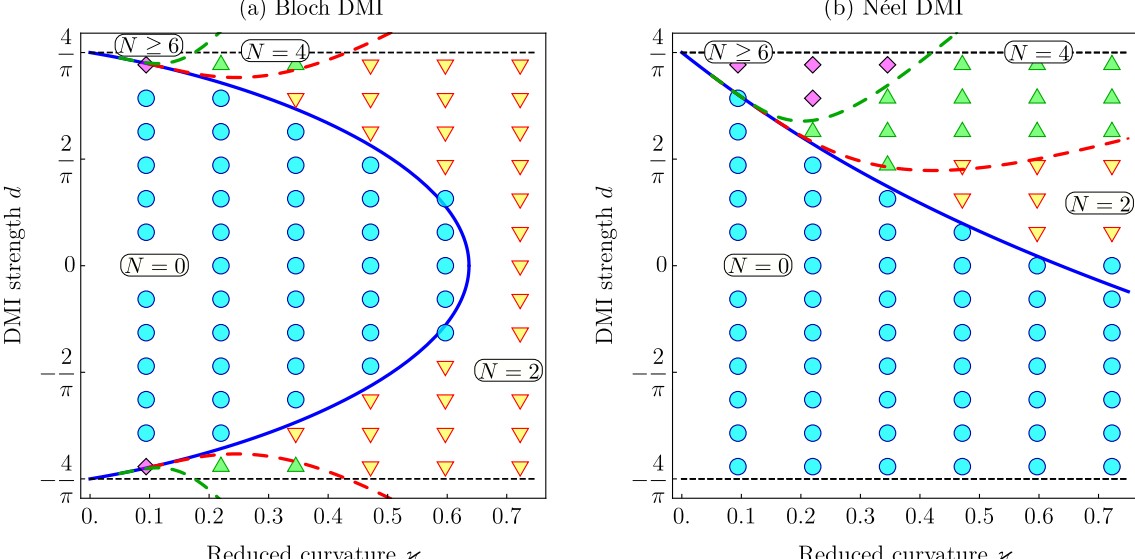

Figure 5: **Equilibrium states in tubular shells with dipole-dipole interaction:** (a) and (b) show phase diagrams of equilibrium states in tubular shell with Bloch and Néel type DMI, respectively. Symbols display the results of spin-lattice simulations: circles – normal (hedgehog) magnetization distribution ($\boldsymbol{m} = \pm\boldsymbol{n}$); other symbols – periodic states (purple diamond correspond to states with $q \geq 3$). Blue solid lines in (a) and (b) are analytical critical lines determined by Eqs. (8) and (12), respectively; dashed lines in (a) and (b) mark transitions between the periodic equilibrium states with different number of DWs, as determined by numerical solution of equations $\mathscr{E}_{\mathrm{B}}^{\mathrm{per}}(q) = \mathscr{E}_{\mathrm{B}}^{\mathrm{per}}(q+1)$ and $\mathscr{E}_{\mathrm{N}}^{\mathrm{per}}(q) = \mathscr{E}_{\mathrm{N}}^{\mathrm{per}}(q+1)$: $q = 1$ corresponds to red dashed line, $q = 2$ – green. Dashed black horizontal lines correspond to critical DMI parameter in a flat systems $d_0 = \pm 4/\pi$.

The dynamical problem is considered as a set of $3N_1N_2$ ordinary differential equations (C.1) with respect to $3N_1N_2$ unknown functions $m_{\boldsymbol{p}}^{\mathrm{x}}(\tau)$, $m_{\boldsymbol{p}}^{\mathrm{y}}(\tau)$, $m_{\boldsymbol{p}}^{\mathrm{z}}(\tau)$. For given initial conditions, the set of time evolution equations (C.1) is integrated numerically using Runge–Kutta method in Python. During the integration process, the condition $|\boldsymbol{m}_{\boldsymbol{p}}(\tau)| = 1$ is controlled.

## C.1 Simulations of tubes without dipole-dipole interaction ($\Lambda = 0$)

We considered cylinders with $N_1 = 300a$ and $N_2 = 900a$, quality factor $Q = 2$ (correspond to $\ell = \ell_{\mathrm{x}}$), the magnetic length $\ell \in [4.5a, 34.5a]$ with $\Delta\ell = 3a$, and DMI constant $d \in [-2, 2]$ with $\Delta d = 0.1$. We simulate numerically Landau–Lifshitz–Gilbert equations (C.1) in the over-damped regime ($\alpha = 0.1$) during a long-time interval $\Delta\tau \gg (\alpha\omega_0)^{-1}$.

We performed a set of simulations for various ranges of magnetic and geometrical parameters. We simulate Eqs. (C.1) as described above for eight different initial states, namely, the normal, $q$-domain walls with $q = \{2, 4, 6, 8, 10\}$, and two random states. The final static state with the lowest energy is considered to be the equilibrium magnetization state. We present simulation data in Figs. 2 and 3 by symbols together with theoretical results (plotted by lines).

## C.2 Simulations of tubes with dipole-dipole interaction ($\Lambda = 1$)

We considered cylinders with $N_1 = 200a$ and $N_2 = 600a$, quality factor $Q = 3$ (correspond to $\ell^{\mathrm{eff}} = \ell_{\mathrm{x}}$), the magnetic length $\ell^{\mathrm{eff}} \in [3a, 23a]$ with $\Delta\ell^{\mathrm{eff}} = 4a$, and DMI constant

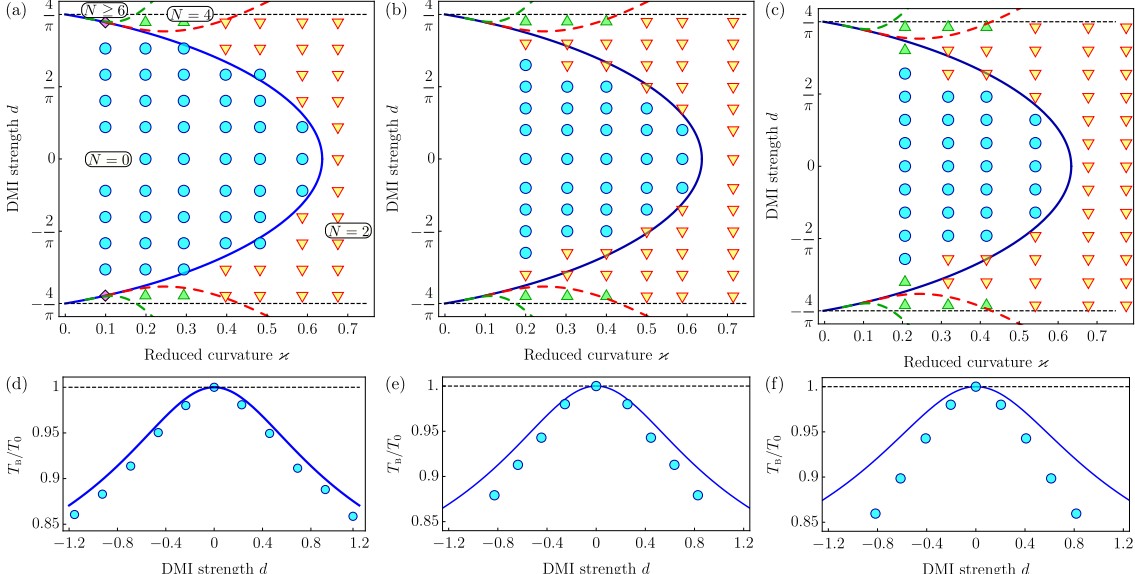

Figure 6: **Phase diagrams of equilibrium states in nanotube with Bloch type DMI.**
(a), (b), and (c) are phase diagrams of equilibrium states in nanotube with Bloch type
DMI: (a) and (d) FeGe epitaxial film with $Q \approx 26.3$; (b) and (e) artificial material
with $Q = 2$; (c) and (f) Pt/Co/AlOx layer structure with $Q \approx 1.71$. In (a)-(c) symbols
display the results of full-scale micromagnetic simulations: circles – normal (hedge-
hog) magnetization distribution ($\boldsymbol{m} = \pm\boldsymbol{n}$); other symbols – periodic states (purple
diamond correspond to states with $q \geq 3$). Blue solid line is analytical critical line
determined by Eq. (8); dashed lines mark transitions between periodic equilibrium
states with different number of DWs, as determined by numerical solution of ener-
gies equality $\mathscr{E}_{\mathrm{B}}^{\mathrm{per}}(q) = \mathscr{E}_{\mathrm{B}}^{\mathrm{per}}(q+1)$: $q = 1$ corresponds to red dashed line, $q = 2$ –
green. Dashed black horizontal lines correspond to critical DMI parameter in a flat
systems $d_0 = \pm 4/\pi$. (d)-(e) are periods $T_{\mathrm{B}}/T_0 = |\cos\psi^{\mathrm{B}}|$ of mgnetization structure
in tube with $\varkappa \approx 0.72$, $\varkappa \approx 0.71$, and $\varkappa \approx 0.78$, respectively.

$d^{\mathrm{eff}} \in [-1.2, 1.2]$ with $\Delta d^{\mathrm{eff}} = 0.2$. The simulations are performed in the same way as de-
scribed in Sec. C.1.

We present simulation data in Fig. 5 by symbols together with theoretical results (plotted
by lines).

# D   Details of full-scale micromagnetic simulations

The micromagnetic simulations were performed with the OOMMF code [57] supplemented
with the extension for the DMI in cubic crystals [58]. Four magnetic interactions were taken
into account, namely exchange, magnetostatic, DMI, and uniaxial anisotropy contributions.
We used the parameters for the epitaxial FeGe film [39,59]: exchange constant $\mathscr{A} = 8.78 \times 10^{-12}$
J/m, saturation magnetization $M_s = 1.1 \times 10^5$ A/m, easy-normal anisotropy $\mathscr{K} = 2 \times 10^5$
J/m³, and DMI constant $\mathscr{D} \in [-1.5, 1.5] \times 10^{-3}$ J/m². These material parameters result
in a quality factor $Q \approx 26.3$ and effective magnetic length $\ell^{\mathrm{eff}} \approx 6.76$ nm. We considered
magnetic nanotubes with fixed length $\tilde{L} = 500$ nm and thickness $h = 4$ nm. The inner ra-
dius of tubes was in the range $R \in [7, 66]$ nm, which results in the dimensionless curvature
$\varkappa = \ell^{\mathrm{eff}}/(R + h/2) \approx [0.1, 0.73]$ (we considered surface between the outer and inner radii).
The mesh size of $0.5 \times 0.5 \times 0.5$ nm³ is used in our simulations.

The simulations are performed in the same way as described in Sec. C.1. Results of numerical simulations are presented in Fig. 2 and Fig. 6(a),(d) by symbols.

### D.1 Full-scale micromagnetic simulations with small quality factor

Additionally we performed simulations for systems with small quality factor:

- We used the following artificial material parameters: exchange constant $\mathscr{A} = 5\pi \times 10^{-12}$ J/m, saturation magnetization $M_s = 5 \times 10^5$ A/m, easy-normal anisotropy $\mathscr{K} = \pi \times 10^5$ J/m$^3$, and DMI constant $\mathscr{D} \in [-1.9, 1.9] \times 10^{-3}$ J/m$^2$. These material parameters result in a quality factor $Q = 2$ and effective magnetic length $\ell^{\text{eff}} = 10$ nm. We considered magnetic nanotubes with fixed length $\tilde{L} = 500$ nm and thickness $h = 4$ nm. The inner radius of tubes was in the range $R \in [12, 48]$ nm, which results in the dimensionless curvature $\varkappa = \ell^{\text{eff}}/(R + h/2) \approx [0.2, 0.71]$ (for the mid-cylinder surface between the outer and inner radii). The mesh size of $1 \times 1 \times 1$ nm$^3$ is used in our simulations.

  Results of numerical simulations are presented in Fig. 6(b),(e) by symbols.

- We used the material parameters of Pt/Co/AlOx layer structure [45]: exchange constant $\mathscr{A} = 1.6 \times 10^{-11}$ J/m, saturation magnetization $M_s = 1.1 \times 10^6$ A/m, easy-normal anisotropy $\mathscr{K} = 1.3 \times 10^6$ J/m$^3$, and DMI constant $\mathscr{D} \in [-3.6, 3.6] \times 10^{-3}$ J/m$^2$. These material parameters result in a quality factor $Q \approx 1.71$ and effective magnetic length $\ell^{\text{eff}} \approx 5.44$ nm. We considered magnetic nanotubes with fixed length $\tilde{L} = 500$ nm and thickness $h = 2$ nm. The inner radius of tubes was in the range $R \in [6, 25]$ nm, which results in the dimensionless curvature $\varkappa = \ell^{\text{eff}}/(R + h/2) \approx [0.21, 0.78]$ (for the mid-cylinder surface between the outer and inner radii). The mesh size of $0.5 \times 0.5 \times 0.5$ nm$^3$ is used in our simulations.

  Results of numerical simulations are presented in Fig. 6(c),(f) by symbols.

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
