# Peer review of "Curvature effects on phase transitions in chiral magnets"

_SciPost Physics, doi:SciPost Phys. 9, 043 (2020)_

## Round 2 · Referee Report · Anonymous (Referee 1) · 2020-7-25

Strengths

1- Elaborate analytical calculations are performed. Their results are compared with those of numerical simulations. The agreement is quantitative, and globally extremely good, the material parameters chosen corresponding to real materials. 2- Non-trivial structures are found, in the form of domain walls running like helices along the nanotube. 3- As experiments on magnetic nanotubes are progressing presently, this work is stimulating.

Weaknesses

For the paper to be fully valuable, some corrections should be made. 1- Some formulas contain mistakes 2 - Both SI and CGS expressions are used 3- Check the English

Report

The paper describes the effect of the curvature-induced Dzyaloshinskii-Moriya interaction (DMI) in magnetic nanotubes, regarding the equilibrium magnetization structures. Elaborate analytical calculations are performed. Their results are compared with those of numerical simulations, either by an atomic spin model, or by using the established public micromagnetic code OOMMF. The agreement is quantitative, and globally extremely good, the material parameters chosen corresponding to real materials. The curvature-induced DMI is superposed to a standard DMI, of the same or of the other symmetry, so as to see their cooperation or competition. In the latter case, non-trivial structures are found, in the form of domain walls running like helices along the nanotube.
As experiments on magnetic nanotubes are progressing presently, this work is stimulating. As far as I know, such results have not been published already.

In the conclusion, it would be good to go beyond the calculation results and explain in simple terms why the effect of curvature is stronger when competing with Neel type DMI.
It would be good also to have some outlook. Indeed, the paper shows that large curvatures are required to get large effects. But when the curvature is too large, the hedgehog structure becomes unstable. Indeed, the \kappa^2 energy of the hedgehog state should be compared to that of the uniform (in 3D space) magnetization, which is 1/2 in the same units, discarding magnetostatic terms. This leads to \kappa < 0.707, close to the value 0.72 often considered in the figures. From this, I reckon that the graphs stop at \kappa=0.72 because, above it, one goes to the uniform magnetization. This would thus be the largest possible curvature.

Requested changes

1) The paper makes a large use of analytical calculations. So the formulas should be carefully checked. I found several mistakes in them, which costed me some time.

  • in (B.1), line 2, the second term should have \sin^2(\phi+\psi)
  • in (B.1), line 3, the cross product \Nabla\theta \times \epsilon should be transformed to a scalar by a dot product with the normal vector n, like in (A.5d)
  • in (B.2), the first equalities for each line do not hold, as numerical factors are lacking. These play no role for the second equalities, as the right-hand side is zero. But they are important if the reader wants to rederive these formulas. So these factors (1/2 in front of dE/dtheta and dE/dphi, 1/(2 kappa) in front of dE/dpsi) should be restored.
  • for (B.4), second line, same comment as for (B.1)
  • for (B.5), same comment as for (B.2)

2) The paper uses sometimes the CGS system, sometimes the SI system. This forces to replicate the column of Table I (with a mistake there: 1 mJ/m^2 is equal to 1 erg/cm^2). I suggest to follow the (not so) modern practice, namely to use SI units throughout.

3) Check the English. Especially for the abstract. The "del operator" mentionned below (A.2) is not a standard term. Why not simply say "gradient" ? The word "whereas" in between (C.2d) and (C.2e) seems to stand for "whether". Ref. [46] should refer to Appendix D.

  • validity: high
  • significance: good
  • originality: high
  • clarity: good
  • formatting: good
  • grammar: good

Author:  Kostiantyn Yershov  on 2020-09-08  [id 947]

(in reply to Report 1 on 2020-07-25)

We thank the Referee for his/her time as well as constructive criticism aimed at improving our manuscript. Following the suggestions, we have improved the paper.

Q1. In the conclusion, it would be good to go beyond the calculation results and explain in simple terms why the effect of curvature is stronger when competing with Neel type DMI. It would be good also to have some outlook. Indeed, the paper shows that large curvatures are required to get large effects. But when the curvature is too large, the hedgehog structure becomes unstable. Indeed, the \kappa^2 energy of the hedgehog state should be compared to that of the uniform (in 3D space) magnetization, which is 1/2 in the same units, discarding magnetostatic terms. This leads to \kappa < 0.707, close to the value 0.72 often considered in the figures. From this, I reckon that the graphs stop at \kappa=0.72 because, above it, one goes to the uniform magnetization. This would thus be the largest possible curvature.

A1. The answer on this comment we would like to split into two parts: - Following Referee’s comment we added into the conclusions the explanation why the curvature induced effects is stronger in the case of the Néel type of DMI:

"The curvature effects are more pronounced for the case of Néel intrinsic DMI because the curvature-induced DMI is usually of the Néel type, thus a direct competition takes place. Note, that for the same reason the Néel skyrmions are more strongly affected by the curvature gradients as compared to the Bloch skyrmions [33] and the DMI-free skyrmions stabilized by curvature are of Néel type [45]." -- page 7 3rd sentence in conclusions. - Indeed, with the increase of the curvature the hedgehog state becomes unstable, and the transition to a nonuniform state, takes place. This transition is denoted by the blue solid line on the phase diagrams shown in Fig. 3. For small enough DMI, the nonuniform state is the two-domain state (q=1) which can be thought of as an onion state of the tube. With the further curvature increase, the two-domain state asymptotically approaches the uniform state with m=m0ˆz. This uniform state has energy 1/2 in the considered units. Note that the other uniform state with m=m0||ˆz has higher energy equal to 1. The comparison of energies shows that there is only one critical curvature value ϰc0.657 (for d=0), which separates the hedgehog and two-domain states. This critical curvature was previously found in Ref. [53]. There are no other critical values. The fact that the phase diagrams in Fig. 3 are limited to ϰ0.72 is just a necessary contingency (the plots must be terminated somewhere).

Q2. The paper makes a large use of analytical calculations. So the formulas should be carefully checked. I found several mistakes in them, which costed me some time. - in (B.1), line 2, the second term should have \sin^2(\phi+\psi) - in (B.1), line 3, the cross product \Nabla\theta \times \epsilon should be transformed to a scalar by a dot product with the normal vector n, like in (A.5d) - in (B.2), the first equalities for each line do not hold, as numerical factors are lacking. These play no role for the second equalities, as the right-hand side is zero. But they are important if the reader wants to rederive these formulas. So these factors (1/2 in front of dE/dtheta and dE/dphi, 1/(2 kappa) in front of dE/dpsi) should be restored. - for (B.4), second line, same comment as for (B.1) - for (B.5), same comment as for (B.2)

A2. We appreciate that the Referee has read our manuscript so carefully and we thank him/her for the spend time. Indeed, in all cases mentioned by the Referee we made misprints. Fortunately, these mistakes are just typos and they do not affect the subsequent calculations. We fixed the misprints in the manuscript.

Q3. The paper uses sometimes the CGS system, sometimes the SI system. This forces to replicate the column of Table I (with a mistake there: 1 mJ/m^2 is equal to 1 erg/cm^2). I suggest to follow the (not so) modern practice, namely to use SI units throughout.

A3. Following the recommendation of the Referee we proceed to SI units. Note that the supplementals are mainly affected, since we use the dimensionless units for the main text.

Q4. Check the English. Especially for the abstract. The "del operator" mentionned below (A.2) is not a standard term. Why not simply say "gradient" ? The word "whereas" in between (C.2d) and (C.2e) seems to stand for "whether". Ref. [46] should refer to Appendix D.

A4. We thanks the Referee for his comment. In the DMI energy, we have a divergence, curl, and gradient. Therefore we used the term ”del operator“. We also revised the text accordingly.

Attachment:

critical_DMI_reply.pdf

---

## Round 2 · Referee Report · Anonymous (Referee 2) · 2020-8-5

Strengths

  1. In their work, it is shown that the curvature changes the DMI strength and new types of domain walls are predicted to appear in the considered system.

  2. The work is very interesting and brings important contributions to the understanding of curvature effects in nanomagnets.

  3. The results are new and innovative.

  4. Curvature effects in magnetic nanoparticles are a hot topic in magnetism researches.

Weaknesses

  1. The presentation of the results needs to be improved.

  2. English needs to be improved.

  3. The bibliography needs to be amended.

Report

The authors study the equilibrium state of ferromagnetic nanotubes with DMI od different symmetries. In their work, it is shown that the curvature changes the DMI strength and new types of domain walls are predicted to appear in the considered system.

The work is very interesting and brings important contributions to the understanding of curvature effects in nanomagnets. However, before being accepted for publication, some issues should be clarified.

Requested changes

  1. Do the authors study the possibility of the solution corresponding to the hedgehog state is a special case of a general solution given by Eq. (5)?

  2. From the analysis of Fig. 3, one can state that there are regions in which Neel DMI and Bloch DMI coexist? If yes, it would be useful to include this discussion in the text.

  3. There are some parts of the text that are confusing. For instance, the presentation of hedgehog and inhomogeneous solutions are presented without proper separation. This fact can bring some difficulties in the understanding of the results. I recommend the authors to perform a revision in the text to better present their results. For instance, there are some parameters that are not presented immediately after appearing in the equations, as the integration constant C.

  4. The text should be revised. There are some problems with English. For instance: “one obtains”, “is reads”, and others.

  5. What do the authors mean with the “simultaneous action of DMI and curvature”?

  6. I call the attention of the authors for some interesting results regarding curvature effects in nanomagnets with DMI. Some of them were developed by authors of this paper: Phys. Rev. B 102, 014432 (2020); https://doi.org/10.1038/s42005-020-0387-2; Phys. Rev. B 102, 024444 (2020); Nanotechnology 31, 125707 (2020); J. Appl. Phys. 108, 033917 (2010); https://doi.org/10.1038/s41598-019-45553-w; and others.

  • validity: high
  • significance: high
  • originality: high
  • clarity: ok
  • formatting: good
  • grammar: reasonable

Author:  Kostiantyn Yershov  on 2020-09-08  [id 946]

(in reply to Report 2 on 2020-08-05)
Category:
question

We thank the Referee for his/her time as well as constructive criticism aimed at improving our manuscript.
Following the suggestions, we have improved the paper.

Q1. Do the authors study the possibility of the solution corresponding to the hedgehog state is a special case of a general solution given by Eq. (5)?

A1. No, formulas (5) and (6) describe the nonuniform multidomain state with q1. The uniform hedgehog state with q=0 we consider separately. In principle, formula (5) can be used for the hedgehog state at the limit case T(C0). However, we believe that such a generalization may confuse the reader.

Q2. From the analysis of Fig. 3, one can state that there are regions in which Neel DMI and Bloch DMI coexist? If yes, it would be useful to include this discussion in the text.

A2. The DMI strength shown in Fig. 3 is the intrinsic DMI (not curvature induced). And in our MS we do not consider the joint action of intrinsic Bloch and Néel DMI. In order to prevent the possible confusion, we denoted it in the caption of Fig. 3:

"... with Bloch and Néel type of intrinsic DMI ..." -- Page 5 caption of Fig. 3

Q3. There are some parts of the text that are confusing. For instance, the presentation of hedgehog and inhomogeneous solutions are presented without proper separation. This fact can bring some difficulties in the understanding of the results. I recommend the authors to perform a revision in the text to better present their results. For instance, there are some parameters that are not presented immediately after appearing in the equations, as the integration constant C.

A3. We thanks the referee for his comment. The homogeneous hedgehog state is trivial, therefore we did not consider it in a subsection. Following the Referee’s recommendation, we highlighted the words “homogeneous” and “inhomogeneous” in the main text. We insert the following text before the formula (5):

"... normal magnetization component of the inhomogeneous state ..." -- page 5 before Eq. (5)

And we add the description of constant C:

"... C is an integration constant ..." -- page 5 after Eq. (5)

Q4. The text should be revised. There are some problems with English. For instance: “one obtains”, “is reads”, and others.

A4. We proofread the text.

Q5. What do the authors mean with the “simultaneous action of DMI and curvature”?

A5. We reformulate it as follows:

"... for the case \varkappa > 0 and d^2 > 0 the DW width decreases as compared to the case ϰ=0 (planar film) or d=0." -- page 6 before formula (7)

And as follows:

"... due to non-zero DMI and curvature the width of the well separated Néel DWs is increased. This behavior is opposite to the case of the Bloch DWs." -- page 7 before formula (11)

Q6. I call the attention of the authors for some interesting results regarding curvature effects in nanomagnets with DMI. Some of them were developed by authors of this paper: Phys. Rev. B 102, 014432 (2020); https://doi.org/10.1038/s42005-020-0387-2; Phys. Rev. B 102, 024444 (2020); Nanotechnology 31, 125707 (2020); J. Appl. Phys. 108, 033917 (2010); https://doi.org/10.1038/s41598-019-45553-w; and others.

A6. We thank the Referee’s for his comment. We added the corresponding citations in the introduction with Refs. [33, 34] and Ref. [23] into the discussion about magnetostatic interaction.

Attachment:

reply.pdf

---

## Editorial Decision

published